# Evaluation of Untargeted Metabolomic and Mycotoxin Profiles in Corn Silage and High-Moisture Corn

**DOI:** 10.3390/toxins17050214

**Published:** 2025-04-24

**Authors:** Marco Lapris, Valentina Novara, Mattia Masseroni, Michela Errico, Gabriele Rocchetti, Antonio Gallo

**Affiliations:** Department of Animal Science, Food and Nutrition, Università Cattolica del Sacro Cuore, Via Emilia Parmense 84, 29122 Piacenza, Italy; marco.lapris@unicatt.it (M.L.); valentina.novara@unicatt.it (V.N.); mattia.masseroni@unicatt.it (M.M.); michela.errico@unicatt.it (M.E.)

**Keywords:** UHPLC-HRMS, screening, feed quality, polyphenols, fungal metabolites, polyamines

## Abstract

Corn silage (CS) and high-moisture corn (HMC) represent fundamental ingredients in ruminant diets; however, their chemical complexity and susceptibility to mycotoxin contamination pose challenges for feed safety and quality assessment. This study applied an innovative approach combining untargeted metabolomics and mycotoxin profiling through ultra-high-performance liquid chromatography–high-resolution mass spectrometry (UHPLC-HRMS) to characterize the chemical profiles of CS (*n* = 19) and HMC (*n* = 13) samples collected from four farms in northern Italy over a period of two years. Fumonisin B1 (FB1) emerged as the most prevalent mycotoxin, with contamination levels significantly higher in HMC than CS, though all the detected levels complied with European Union (EU) guidance limits. Untargeted metabolomics distinguished CS and HMC based on their metabolic signatures: polyamines, amino acids, peptides, and phenolic acids typified CS, while HMC was primarily characterized by flavonoids and mycotoxins. Geographical origin significantly influenced both mycotoxin patterns and metabolite profiles, while the sampling season showed no significant impact. This study highlights the complementary value of metabolomics and mycotoxin screening to assess feed quality, identify biomarkers, and unravel the link between fungal contamination and biochemical composition, offering a robust strategy to support feed safety management in livestock production.

## 1. Introduction

Corn silage (CS) and high-moisture corn (HMC) are fundamental components of ruminant diets due to their high energy content and digestibility, supporting milk and meat production in dairy and beef systems [1,2]. However, the quality and safety of these feedstuffs are often compromised by fungal contamination, leading to the potential presence of mycotoxins. Mycotoxins are secondary metabolites produced by various fungal species, particularly those belonging to the genera *Fusarium*, *Aspergillus*, and *Penicillium* [3]. Their occurrence in feed represents a significant concern for animal health and productivity, with potential carry-over effects on animal-derived products and implications for food safety [4].

While regulated mycotoxins such as aflatoxins, deoxynivalenol, zearalenone, fumonisins, and ochratoxins have been extensively monitored, increasing attention is being paid to emerging mycotoxins and masked forms, whose toxicological effects and occurrence patterns remain less understood [5]. Concurrently, the metabolomic composition of silages and fermented feedstuffs is receiving growing interest [6,7]. The chemical profile of these matrices is influenced by several factors, including plant genotype, agronomic practices, ensiling conditions, microbial activity, and environmental variables [8].

Untargeted metabolomics, a powerful tool relying on high-resolution mass spectrometry (HRMS), enables the comprehensive characterization of metabolites in complex biological samples, providing insights into both nutritional quality and the biochemical consequences of fungal contamination [9]. Combining untargeted metabolomics with multi-screening mycotoxin analysis can yield a more holistic evaluation of feed quality, enabling the detection of known and unknown bioactive compounds that may affect animal performance and health [10]. This integrated approach is particularly relevant in regions characterized by climate variability, as temperature and humidity fluctuations can influence fungal growth and mycotoxin biosynthesis, potentially exacerbating contamination risks [11].

Despite the importance of CS and HMC in ruminant feeding, few studies have investigated their metabolomic profiles alongside a comprehensive assessment of mycotoxin contamination. In particular, consumption of mycotoxin-contaminated silages can significantly alter the metabolic profiles of animals [11,12], affecting mainly energy metabolism, amino acid metabolism, liver and gut health biomarkers, and neurotransmitter and hormonal disruptions. Moreover, limited information is available regarding the influence of geographical and seasonal variability on the chemical composition of these feeds. Understanding the interplay between mycotoxins and the metabolic landscape of feed ingredients can support the development of targeted mitigation strategies and improve risk management practices in livestock production systems.

Therefore, starting from this background, this research survey aimed to characterize the untargeted metabolomic profiles and mycotoxin contamination patterns of CS and HMC samples collected from dairy farms in northern Italy. An approach using advanced ultra-high-performance liquid chromatography coupled with high-resolution mass spectrometry (UHPLC-HRMS) was applied, enabling the simultaneous detection of both regulated and emerging mycotoxins, as well as the comprehensive profiling of endogenous metabolites. Multivariate statistical analyses were employed to investigate the effects of feed type and geographical area on the chemical composition of these feed matrices. This work provides novel insights into feed quality assessment, emphasizing the value of integrating metabolomics with mycotoxin screening to support the safety and efficiency of ruminant nutrition.

## 2. Results

### 2.1. Chemical and Fermentative Parameters of CS and HMC Samples

The chemical composition and microbial counts, together with other fermentative parameters, of CS and HMC are presented on a dry matter (DM) basis in Appendix A. The CS group was characterized by 7.9% protein, 2.5% fat, 4.3% ash, 32.6% starch, 39.7% neutral detergent fiber (NDF), 24.5% acid detergent fiber (ADF), and 3.3% acid detergent lignin (ADL). In contrast, the HMC group showed 8.7% protein, 3.4% fat, 1.7% ash, 61.3% starch, 14.1% NDF, 6.3% ADF, and 1.3% ADL (Appendix A). Regarding microbial counts, LAB were detected at higher levels in HMC compared to CS, with average counts of 7.25 logCFU/g in HMC versus 5.13 logCFU/g in CS. Despite the lower LAB counts in CS, lactic acid concentration was higher in CS than in HMC, reaching 4.9% DM in CS compared to 2.3% DM in HMC. Additionally, the optimal fermentative quality of the samples was further confirmed by the absence of volatile organic compounds (VOCs) typically associated with Clostridia fermentation, such as isovaleric acid, isobutyric acid, valeric acid, and butyric acid (Appendix A). From an overall perspective, the CS samples were characterized by a higher concentration of total alcohols compared to HMC, averaging 1.23% DM versus 0.48% DM, respectively. Additionally, propylene glycol levels were consistently higher in CS than in HMC (Appendix A), although considerable variability was observed within the CS group, with concentrations ranging from 0.2% to 17.2% DM. Lastly, acetic acid was detected in both the CS and HMC samples under investigation (Appendix A) and was higher in the CS group (3.1% DM) than the HMC group (0.7% DM).

### 2.2. Co-Occurrence of Regulated and Emerging Mycotoxins in CS and HMC Samples

In this study, feed samples were screened using an UHPLC-HRMS approach. The results (Table 1) revealed that almost all the samples were positive for fumonisins B1, B2, and B3 (FB1, FB2+FB3), zearalenone (ZEN), deoxynivalenol (DON), beauvericin (BEA), and fusaric acid (FA). In contrast, aflatoxins (AFB1, AFB2, AFG1, AFG2), T-2, HT-2, and OTA were absent or detected at very low levels, with all values falling below the lowest point of the calibration curve in the solvent (i.e., 1 μg/kg) for both the CS and HMC samples. A matrix effect (ME%) for regulated mycotoxins in CS was evaluated for each mycotoxin by comparing the slopes of the calibration curves prepared in the solvent and in the silage matrix. The calculated ME values were as follows: AFB1 = −59.6%, FB1 = +3.6%, FB2 = −23.0%, DON = −0.3%, and ZEA = −0.2%. A detailed sample legend, including information on feed type, geographical area, analytical method validation parameters, and sampling time for each analyzed sample, is provided in Appendix A.

Focusing on regulated mycotoxins, FB1 contamination was significantly higher in HMC compared to CS (*p* < 0.05). The average FB1 concentration in CS was 533.2 μg/kg, although some samples (e.g., samples 45, 55, and 58) reached levels between 1000 and 2000 μg/kg. In the HMC samples, the average FB1 level was nearly three times higher, with samples 39, 42, and 44 exceeding 3000 μg/kg. Similarly, significant differences (*p* < 0.05) were detected for FB2+FB3. The CS samples showed a combined average contamination of 258.5 μg/kg, with most values ranging between 150 and 250 μg/kg, although samples 40, 52, 55, and 58 presented higher levels (500–800 μg/kg). In HMC, the average FB2 and FB3 contamination was approximately 500 μg/kg, with samples 39, 42, and 44 showing the highest levels (1100–1200 μg/kg). ZEN contamination followed a similar pattern (although not significant; *p* > 0.05), with CS samples averaging 23.1 μg/kg, though sample 37 exhibited a high level exceeding 100 μg/kg. In HMC, ZEN contamination showed an average level of 78.2 μg/kg; samples 30 and 54 displayed notably higher concentrations, surpassing 200 μg/kg and 400 μg/kg, respectively. DON was present in both feed types (*p* > 0.05), with average contamination in CS of 367.2 μg/kg (ranging from 14.7 to 515.2 μg/kg). However, samples 34, 37, 52, and 59 displayed markedly higher levels, reaching 684, 1104, 956.8, and 736 μg/kg, respectively. In HMC, the DON levels were generally lower, with an average of 259.2 μg/kg, and most samples fell between 10 and 270 μg/kg. Nonetheless, two samples (30 and 54) showed high contamination levels (1658 μg/kg and 706.5 μg/kg, respectively). Finally, the analysis of emerging mycotoxins indicated comparable average contamination levels (*p* > 0.05) in both CS and HMC for BEA, being 24.7 μg/kg and 18.3 μg/kg, respectively. The BEA concentrations in CS were generally below 50 μg/kg, with the exception of sample 58, which showed a notably high value of 172.7 μg/kg. In HMC, the highest level was recorded in sample 44 (41.2 μg/kg). The FA levels displayed no significant differences between CS and HMC; however, a great variability was observed, ranging from 200 to over 3000 μg/kg in CS (sample 58) and from 300 to nearly 5000 μg/kg in HMC (sample 42).

### 2.3. Untargeted Metabolomic Profiles of CS and HMC Samples

The untargeted metabolomics approach based on UHPLC-HRMS allowed the annotation of almost 7000 mass features that were used to perform the multivariate statistical analyses. In particular, both the unsupervised and supervised methods were used to separate the CS and HMC samples according to the measured chemical profiles. The output of the hierarchical clustering analysis (HCA; heat map) is provided in Figure 1A, showing a clear discrimination between CS and HMC when considering the Log_2_ variations of each single mass feature. Looking at the CS group (red color), it was evident that some clusters of metabolites were completely absent in HMC (green color), and vice versa. The same output was obtained for the orthogonal projection to latent structure (OPLS)–discriminant analysis (DA) score plot (Figure 1B), which outlined on the first latent vector a clear separation between the CS (green circles) and HMC (blue squares) samples, with a goodness of prediction (cumulative Q^2^) equal to 0.986, thus confirming the robustness of the annotated mass features to discriminate the two sample groups. Also, a second discrimination was outlined for the HMC group on the second latent vector, by distinguishing between HMC samples made with only corn grain or the whole plant.

The most significant metabolites for predicting the chemical differences between the HMC and CS samples were extrapolated using the VIP (variable’s importance in projection) selection method, setting as a prediction threshold a value higher than 1 (VIP > 1). Overall, 2007 mass features were found as potential biomarkers between CS and HMC; however, only 123 metabolites (45 for HMC and 78 for CS samples) were positively identified against the comprehensive database in the ESI ionization mode available in the MS-DIAL software (version 4.90) (Appendix A) and possessed a significant variation in the pairwise comparison (resulting from a volcano plot analysis; Appendix A). The most represented chemical classes by the annotated metabolites were amino acids and derivatives, followed by fatty acids and derivatives, phenolic compounds and metabolites, carbohydrates and sugar derivatives, purines and pyrimidines, organic acids, peptides, and other metabolites (such as polyamines, terpenoids, mycotoxin metabolites, and plant hormones). An overview of the most represented discriminant classes for both CS and HMC is reported in the pie charts of Figure 2A and Figure 2B, respectively. Polyphenols followed by amino acids and derivatives were the most up-accumulated and discriminant chemical classes found in CS, recording mostly phenolic acids (both hydroxybenzoics and hydroxycinnamics) and stilbenes (such as resveratrol and derivatives) (Appendix A). However, the most discriminating ability was found for genipin (VIP score = 1.980; Log_2_FC = 4.036) and apigenin (VIP score = 1.678; Log_2_FC = 3.648) belonging to the iridoid and flavone classes, respectively. Regarding the amino acids and derivatives in the CS samples, the highest discriminating ability was found for the compound acetylcarnitine (VIP score = 1.740; Log_2_FC = 3.289), followed by cycloserine (VIP score = 1.721; Log_2_FC = 3.115). Similarly, among the metabolites possessing the highest prediction ability in HMC, we found polyphenols (40%), followed by amino acids and derivatives (19%) (Figure 2B). However, by inspecting the list of significantly up-accumulated phenolics, we found mainly flavonoids belonging to several sub-classes, such as flavonols and flavones. Accordingly, the most significant prediction ability was found for the flavonol myricetin-3-rutinoside, showing a definitely high up-accumulation trend when compared with CS (Log_2_FC = 9.794). Another potential phenolic biomarker of HMC was represented by isovitexin (a flavonoid C-glycoside), showing a Log_2_FC = 12.725 for the pairwise comparison of HMC vs. CS (Appendix A). The entire list of discriminant metabolites for both CS and HMC is available as Appendix A.

Among other metabolites discriminating the CS vs. HMC samples, we found polyamines (6%; Figure 2A). This class of compounds was outlined as exclusively discriminant for the CS group. As shown in Figure 3, significant variations were measured for tyramine (VIP score = 1.383; Log_2_FC = 1.657), putrescine (VIP score = 1.314; Log_2_FC = 2.099), histamine (VIP score = 1.999; Log_2_FC = 4.592), and cadaverine (VIP score = 1.196; Log_2_FC = 1.517).

### 2.4. Impact of Geographical Origin and Sampling Time on the Chemical Profiles of CS and HMC Samples

The AMOPLS analysis (Table 2) highlighted that geographical origin was the most influential factor affecting the chemical profiles of both the CS and HMC samples, with a relative contribution of 20.7% for CS and 29.5% for HMC. This result suggests that farm location, i.e., the terroir effect, plays a key role in shaping the metabolic fingerprint of the analyzed feed matrices. On the other hand, sampling time (season) had a minimal impact on chemical variability (5.4% for CS and 7.8% for HMC), with no statistical significance detected for this factor. The interaction between origin and sampling time explained 10.3% of the variability in CS and 9.6% in HMC, but its significance was also limited (*p* > 0.05).

The OPLS-DA score plots (Figure 4) further supported these findings, as samples were distinctly clustered based on their farm of origin, regardless of the sampling time (winter vs. summer). This clear separation across farms suggests that local agronomic practices, soil characteristics, and microclimatic conditions likely exert a stronger influence on feed chemical composition than seasonal variation, thus stimulating the biosynthesis of secondary metabolites, such as polyphenols. This was particularly evident for both the CS and HMC samples, with flavonoids–glycosides, phenolic acids, and terpenoids emerging as key biomarkers of geographical origin (Appendix A). Therefore, multivariate statistics (both OPLS-DA and AMOPLS) showed the importance of terroir in determining the chemical signature of feed ingredients, underscoring the potential of untargeted metabolomics for origin authentication and quality evaluation in corn-based feedstuffs.

## 3. Discussion

The present study provided a comprehensive evaluation of the chemical composition of CS and HMC, two widely used feed ingredients in ruminant nutrition, through the combination of untargeted metabolomics and mycotoxin profiling.

To achieve optimal silage and maximize dry matter (DM) and energy conservation, lactic acid bacteria (LAB) play a key role, utilizing available water-soluble carbohydrates for lactic acid production [13]. Lactic acid is the main agent responsible for lowering pH, contributing significantly to silage stabilization and preservation [1]. In the present study, lactic acid was detected in all the samples analyzed (Appendix A). The obtained values were strictly in agreement with those previously reported by Kung et al. [1]. The differences in chemical composition and microbial counts between CS and HMC reflect their distinct fermentation processes and plant matrix characteristics. The higher protein, fat, and starch contents observed in HMC align with its predominantly grain-based composition, while the elevated fiber, NDF, ADF, and ADL levels in CS (Appendix A) are expected, given its whole-plant nature, which includes leaves, stems, and cobs. The higher production of lactic acid in the CS group could be attributed to several factors. Firstly, the higher fiber content in CS likely provided a more diverse and sustained substrate for microbial fermentation, promoting lactic acid accumulation over time. Additionally, the slower fermentation dynamics in CS may have allowed a more gradual and prolonged lactic acid production compared to the rapid starch fermentation typical of HMC. Another key factor could be the dominance of different LAB species in the two matrices. HMC tends to favor fast-growing homofermentative LAB that rapidly convert simple sugars into lactic acid, leading to early acidification but potentially limiting the final lactic acid concentration. In contrast, CS may support a more complex microbial community, including heterofermentative LAB like *Lactobacillus buchneri*, which not only produce lactic acid but also contribute to the formation of secondary metabolites such as acetic acid and propylene glycol. The higher lactic acid concentration in CS, despite lower LAB counts, suggests a more efficient preservation process, as lactic acid is crucial for lowering pH and inhibiting spoilage microorganisms. Conversely, the rapid fermentation and higher LAB counts in HMC, combined with lower lactic acid accumulation, may reflect a shorter, less stable fermentation phase, potentially making HMC more susceptible to secondary fermentation if storage conditions are not well controlled. There was also no evidence of organic compounds from *Clostridia* fermentation, which are known to be associated with lower acidification and reduced silage preservability [14]. Interestingly, we positively recorded in almost all the CS samples (Appendix A) the compound propylene glycol (1,2-propanediol), likely reflecting microbial activity during secondary fermentation, particularly that involving heterofermentative LAB, such as *Lactobacillus buchneri*. This metabolite is produced through the conversion of lactic acid into 1,2-propanediol and subsequently acetic acid, contributing to silage stabilization by lowering pH and inhibiting spoilage microorganisms [15]. The absence or lower levels of propylene glycol in HMC align with a reduced microbial fermentation. From a nutritional standpoint, propylene glycol serves as a glucogenic precursor for ruminants, thus supporting gluconeogenesis and energy balance, particularly for dairy cows during early lactation [15].

The mycotoxin profile detected here (Table 1) highlighted a greater contamination level in HMC compared to CS, with FB1 being the most prominent contaminant in HMC. This observation aligns with previous studies indicating that corn kernels are particularly susceptible to fumonisin contamination, especially under pre-harvest conditions conducive to *Fusarium* proliferation. Conversely, the ensiling process and the LAB activity involved in CS production may have contributed to mitigating fungal development and reducing mycotoxin accumulation. Despite the general presence of mycotoxins in both matrices, none of the detected levels exceeded the European Union (EU) guidance values for animal feed (Commission Recommendation 2006/576/EC), ensuring a satisfactory safety profile of the analyzed feeds (Table 1). Overall, these quantitative results demonstrate that mycotoxin contamination is widespread in both CS and HMC, with fumonisins being the dominant mycotoxins, especially in HMC samples, which exhibited notably higher levels compared to CS. Regarding method validation, our results indicate ionization suppression for AFB1 (consistent with a 62.7% recovery in silage extract; Appendix A), moderate suppression for FB2, and negligible matrix effects for FB1, DON, and ZEA. Therefore, the signal suppression observed for AFB1 underscores the importance of matrix-matched calibration for reliable quantification in silage samples. Overall, the low matrix effects observed for most of the analytes confirm the suitability of the method for quantitative analysis in this complex feed matrix.

While emerging mycotoxins such as BEA and FA were present at lower levels, their variability across samples suggests the need for further attention in feed quality monitoring. In particular, BEA and FA are emerging mycotoxins, mainly produced by *Fusarium* species. However, to date, they are not regulated at the European level under Regulation (EU) 2023/915, nor are they included in mandatory monitoring programs or listed among the substances requiring validation according to the provisions of Regulation (EU) 2021/808 or the SANTE/11312/2021 guidelines. For this reason, our analytical method was formally validated only for the regulated mycotoxins that are toxicologically and legislatively most relevant for silage, namely AFB1, FB1, DON, and ZEA (Appendix A). Nevertheless, the potential presence of BEA and FA in the silage samples was monitored in an exploratory and non-quantitative manner, with the aim of providing additional information on the mycotoxin profile and highlighting the potential of the omics-based approach.

A previous comprehensive survey by Weaver et al. [16] on the co-occurrence of 35 mycotoxins in corn grain and CS in the United States revealed over seven years, showed that FA was the most frequently detected mycotoxin in 78.1% and 93.8% of corn grains and silages, respectively. In particular, these authors outlined that some of the more prevalent mycotoxins in these matrices were those that are scarcely analyzed by routine analyses, such as FA and DON metabolites. Therefore, assessment of multiple mycotoxins should be considered when developing management programs. Overall, it is also important to outline that the contamination of silage can be affected by several factors across the whole farm environment, from the field to the bunk. Some of these factors can be related to forage species, stage of maturity or moisture content at harvest time, ensiling processes, storage structure, use of silage additives or oxygen barrier films, feed-out methods, and bunk management [17].

As recently pointed out by Adeniji et al. [18], metabolomics has the potential to be a crucial tool for unraveling the biological pathways involved in the cross-talk between toxigenic fungi and their host plant or between toxigenic fungi and the soil/grain microflora. Metabolomics is therefore a relevant strategy for identifying key biochemical factors responsible for a modulation of the production of mycotoxins. UHPLC-HRMS revealed substantial differences in the metabolite profiles between CS and HMC (Figure 1 and Figure 2). Polyamines, such as putrescine and spermidine, were particularly enriched in CS, reflecting microbial fermentation processes and microbial metabolic activity during ensiling (Figure 3). Biogenic amines arise from decarboxylation of amino acids, based on the action of either plant enzymes or microbial enzymes of various species of LAB (*Lactobacillus*, *Pediococcus*, and *Streptococcus*) and species of the genera *Clostridia*, *Bacillus*, *Klebsiella*, *Escherichia*, *Pseudomonas*, *Citrobacter*, *Proteus*, *Salmonella*, *Shigella*, and *Photobacterium*. Determining amine concentrations in silage may help to indicate undesirable changes in forages and could prevent possible toxicity for livestock [19]. However, to date, amine analyses have not been included in the standard chemical analyses of forages. These compounds, though not regulated, are increasingly recognized as indicators of silage quality and microbial stability [20]. An intriguing biochemical relationship emerged from the observed high levels of polyamines in CS and the concomitantly lower contamination by fumonisins compared to HMC. Polyamines, such as putrescine and spermidine, can be also associated with antifungal properties and plant defense mechanisms [20]. Polyamines can inhibit fungal growth or reduce mycotoxin biosynthesis by altering cell membrane stability, interfering with fungal metabolism, or enhancing plant resistance pathways. A previous study on corn has also pointed out an implication of a variety of polyamines in response to *F. graminearum*, such as cadaverine [21]. During ensiling, the production of polyamines by LAB and other microbes might create a less favorable environment for toxigenic fungi, contributing to the lower levels of FBs detected in CS. This potential protective role of polyamines aligns with previous evidence highlighting the complex interactions between microbial metabolites and fungal contamination during silage fermentation. Future research should further explore the mechanistic role of polyamines in mycotoxin mitigation and their possible use as biomarkers of silage safety and quality.

The CS samples were also characterized by higher levels of amino acids, peptides, and phenolic acids, which are known to be associated with plant metabolism and ensiling biochemistry. In contrast, HMC samples were discriminated by an abundance of flavonoids and mycotoxins. Interestingly, most of the *Fusarium* head blight (FHB)-resistance metabolites evidenced so far derive from three plant metabolic pathways—the shikimate, acetate–mevalonate, and methylerythritol pathways—and belong to different groups, including flavonoid phenylpropanoids, non-flavonoid phenylpropanoids, fatty acids, glycerophospholipids, terpenoids, amino acids, amines, polyamines, and carbohydrates [9]. It has been proposed that these metabolites are involved in a plant defense system triggered to counteract toxigenic fungal pathogens through their involvement in several key mechanisms, including cell wall reinforcement, with the deposition of lignin and/or callose and the specific induction of defense signaling pathways [22,23]. Moreover, several of these resistance-related metabolites have been shown to display antifungal properties and, for a limited number of them, capacities to interfere with the production of mycotoxins [20]. In this work, most of the discriminating metabolites between CS and HMC were medium-chain and long-chain fatty acids, amino acids and derivatives, terpenoids, polyamines, organic acids, and carbohydrates (Figure 2). Among the significant and discriminant up-accumulated compounds in CS, we found the plant hormone dihydrojasmonic acid (VIP score = 1.669; Log_2_FC = 2.636; *p*-value = 1.72 × 10^−16^). Jasmonates, including jasmonic acid and its derivatives, are well-known phytohormones that play a pivotal role in plant defense responses against biotic and abiotic stresses. These compounds are part of the plant’s systemic acquired resistance and are involved in signaling pathways that activate the production of secondary metabolites, such as phenolic acids and alkaloids, with antimicrobial and antifungal properties [24]. Jasmonates have been shown to enhance plant defenses against *Fusarium* spp. and other fungal pathogens by modulating the synthesis of defense-related proteins and increasing the production of protective metabolites [25]. Therefore, the higher levels of dihydrojasmonic acid observed in CS may reflect a stress response induced during the ensiling process (i.e., because of post-harvest metabolism) or pre-harvest conditions, contributing to the observed reduction in fumonisin contamination. This suggests that jasmonate signaling could play a role in the natural defense mechanisms of CS, potentially enhancing resistance to fungal colonization and mycotoxin production [26]. Another difference clearly emerging when comparing CS and HMC metabolomic profiles was represented by the distribution of purines and pyrimidines (Figure 2). Among the discriminant purines in CS, we found allantoin (VIP score = 1.225). The higher presence of allantoin in CS compared to HMC likely reflects purine catabolism and the nitrogen recycling pathways activated in plant tissues during the ensiling process. Allantoin is a ureide compound derived from purine degradation, often accumulating in response to oxidative stress and nitrogen remobilization in plants [27]. The ensiling process involves cutting and fermenting the whole plant, which can trigger stress-related metabolic adjustments, including purine breakdown to allantoin.

Regarding polyphenols, there is a notably large body of evidence that supports the inhibitory activities of cinnamic acid derivatives towards the biosynthesis of mycotoxins, including DON, but also type A trichothecenes, fumonisins, ochratoxin, and aflatoxin [28]. We found a significant up-accumulation of shikimic acid (VIP score = 1.144; Log_2_FC = 1.341; *p*-value = 1.18 × 10^−15^) and several hydroxycinnamic and hydroxybenzoic acids, e.g., 3,4-dimethoxycinnamic acid, 3,4,5-trimethoxycinnamic acid, hydrocinnamic acid, and vanillic acid, in CS samples. Another relevant biochemical aspect highlighted by this study is the up-accumulation of phenolic acids in CS, which coincided with the lower fumonisin contamination compared to HMC. Phenolic acids are well-known plant secondary metabolites with antimicrobial, antifungal, and antioxidant properties. In the context of ensiling, the release and accumulation of phenolic acids may occur due to cell wall degradation and microbial activity. Their presence could contribute to a protective biochemical environment that limits fungal development and mycotoxin biosynthesis during silage fermentation. This aligns with the observation that CS samples exhibited both higher levels of phenolic acids and lower fumonisin contamination. Finally, another key up-accumulated class of compounds in CS was represented by organic acids, including malate, malonate, and galactarate (a product of pectin and hemicellulose degradation) (Figure 2 and Appendix A). The up-accumulation of organic acids in CS compared to HMC can be attributed to plant metabolic processes and microbial fermentation during ensiling. Malic acid is a central intermediate in the tricarboxylic acid (TCA) cycle and is involved in C4 photosynthesis in maize [29]; it also serves as a substrate for malolactic fermentation, a microbial process contributing to silage acidification and stabilization. Malonate is a product of malonyl-CoA metabolism, associated with fatty acid biosynthesis and microbial activity [30], and may also play a role in plant stress responses. Galactarate (mucic acid) is a sugar acid resulting from the oxidative degradation of galactose, likely from pectins and hemicelluloses in plant cell walls [31]. Its accumulation reflects cell wall breakdown during ensiling and microbial carbohydrate metabolism. Collectively, these organic acids contribute to the acidification of silage, creating an unfavorable environment for fungal growth and potentially explaining the lower fumonisin levels in CS. The acidic conditions, combined with other plant and microbial metabolites and likely contribute to the suppression of *Fusarium* proliferation and mycotoxin biosynthesis during silage fermentation. Finally, looking at the discriminant metabolites of CS vs. HMC, metabolomics outlined a key up-accumulation of tryptophan and indole-derivatives, such as indole-3-acetyl-L-phenylalanine and indole-3-acetic acid (Appendix A). As a general consideration, the involvement of aromatic amino acids in resistance against DON-producing *Fusarium* has been directly related with their role as precursors for a wide range of secondary metabolites that play a pivotal role in plant defense against biotic stresses (such as phenolic compounds). In addition, the catabolism of tryptophan leads to many indole-containing secondary metabolites, such as auxins, glucosinolates, and terpenoids, i.e., three classes of compounds largely documented for their implication in plant–pathogen interactions [32].

An additional outcome from this omics survey was the impact of geographical area on both mycotoxin contamination and metabolite composition, as revealed by multivariate statistical analysis (Table 2 and Figure 4). Samples from different farms displayed distinct chemical signatures, with CS being discriminated by 35 VIP metabolites (Appendix A), while 37 VIP metabolites were outlined as the key biomarkers of HMC (Appendix A). Among the most represented classes of compounds in terms of geographical area, we found flavonoids, phenolic acids, and terpenoids. Interestingly, sampling season (winter vs. summer) did not significantly affect the compositional variability, suggesting that the participating farms adopted effective management practices to maintain feed quality across the years (2022–2023). This evidence underscores the importance of terroir in determining feed characteristics and the potential of omics technologies to track the geographical origin of feed materials. To summarize, the integration of mycotoxin screening and metabolomic profiling represents an innovative strategy for holistic feed quality assessment. The associations observed between certain mycotoxins and plant- or microbe-derived metabolites suggest that fungal contamination and metabolic responses are tightly intertwined in corn-based feeds.

## 4. Conclusions

This study applied an integrated approach combining untargeted metabolomics and mycotoxin profiling through UHPLC-HRMS to evaluate the chemical profiles of CS and HMC, two key ingredients in ruminant nutrition. The results highlighted that the HMC samples exhibited higher mycotoxin contamination, particularly fumonisins, compared to the CS samples, though all levels complied with EU guidance values. Untargeted metabolomics enabled the differentiation of the CS and HMC samples, revealing polyamines, amino acids, peptides, organic acids, and phenolic acids as key discriminant metabolites in CS, whereas flavonoids and mycotoxins were the main markers of HMC. Additionally, geographical origin significantly influenced feed chemical composition, while sampling time (winter vs. summer season) had a negligible effect, suggesting consistent feed quality management across seasons. These findings demonstrate the power of omics technologies in feed analysis, offering insights into the relationship between metabolite patterns, mycotoxin contamination, and geographical origin. This approach provides a robust strategy to enhance feed safety, traceability, and quality evaluation, contributing to sustainable livestock production.

## 5. Materials and Methods

### 5.1. Collection and Characterization of Feed Samples

Thirty-two feed samples, comprising CS (*n* = 19) and HMC (*n* = 13), were collected from four farms in northern Italy between 2022 and the end of 2023. Specifically, samples were obtained from “Farm 3” and “Farm 4” in Piedmont (Cuneo), “Farm 2” in Lombardy (Brescia), and “Farm 1” in Veneto (Vicenza). The experimental plan was designed to assess the effects of feed type, geographical origin, and sampling season. Sampling was conducted four to five times per farm, covering both summer and winter seasons. However, with regard to the HMC samples, Farms 2 and 4 contributed only during the winter season.

For the sampling procedure, about 2 kg of CS and HMC (on a wet basis) were collected from four random zones of the feed-out face of horizontal bunker silos, using a hand core drill that took samples that were up to about 40 cm deep from the bunker face. Silage samples that were collected were then split into two homogeneous subsamples of about 1 kg of fresh matter, one for near-infrared (NIR) analysis and the other one for fermentative parameter evaluation. The subsample designated for NIR assessment was dried in a forced-air oven (65 °C, 48 h) for DM determination according to AOAC (1995) [33]. Then, dried samples were ground to pass a 1 mm screen and were characterized by Foss (Hilleroed, Denmark) NIR systems DS3 spectrophotometer equipped with a monochromator, scanning over the wavelength range 400 e 2500 nm every 0.5 nm. The calibrations used to obtain forage characterizations were produced by Foss (Hilleroed, Denmark), and the samples were analyzed for ash, protein, fiber, neutral detergent fiber (NDF), acid detergent fiber (ADF), acid detergent lignin (ADL), calcium, phosphorus, starch, and fat. The other subsample was analyzed as fresh for fermentative parameters, including volatile fatty acids (VFAs), lactic acid, volatile organic compounds (VOCs), including aldehydes, alcohols, ketones, or esters, ammonia, and pH. To quantify VOC or VFA, the extracted solution was prepared and then injected into a gas chromatographic–flame ionization detector (GC/FID) system, as described by Sigolo et al. [13]. Additionally, lactic acid was determined using high liquid performance chromatography (HPLC) after two dilutions with distilled water for lactic acid, as reported by Gallo et al. [19]. Finally, all the samples were ground to a particle size of 0.5 mm to facilitate the extraction of mycotoxins and other metabolites.

### 5.2. Extraction Protocol of Mycotoxins and Small Metabolites

The collected CS and HMC samples (2 g) were added to a volume of 16 mL of the extraction solution, consisting of acetonitrile/water/glacial acetic acid (73.75/25/1.25, *v*/*v*/*v*) in 50 mL falcon tubes. The samples were then homogenized using a Polytron system at maximum power and then centrifuged (5500 rpm), setting a temperature of 4 °C for 15 min. Finally, the extracts were incubated overnight at −20 °C and then filtered using 0.22 μm RC syringe filters in vials for the UHPLC-HRMS analysis.

### 5.3. Targeted and Untargeted Analyses Based on UHPLC-HRMS

The quantification of regulated and emerging mycotoxins is performed using a UHPLC instrument coupled with a Q-Exactive Focus Orbitrap mass spectrometer (Thermo Fisher Scientific, Waltham, MA, USA). The UHPLC system consists of a degassing system, a quaternary UHPLC pump, an autosampler device, and a thermostatically controlled Thermo Scientific™ Hypersil GOLD™ aQ (100 × 2.1 mm, 1.9 μm) held at 35 °C. The mobile phase consists of (A) water with 0.1% formic acid, 2% methanol, containing 5 mM ammonium formate, and (B) methanol with 0.1% formic acid, 2% water, and containing 5 mM ammonium formate. A gradient elution program is applied as follows: an initial 0% B held for 0.5 min, increased to 100% B over 7.5 min, and held for 0.5 min. Then, the gradient decreases to 0% B over 6 min to re-equilibrate the instrument for a total run time of 15 min. The flow rate is 0.3 mL/min, while the injection volume is 3 μL. Detection is performed using a Q-Exactive Focus Orbitrap mass spectrometer, considering two biological replicates for each sample. The mass spectrometer works in both the positive and negative ion modes, by setting 2 scan events: full ion MS and parallel reaction monitoring (PRM) for targeted fragmentation. Full-scan data are acquired at a resolving power of 70,000 for full width at half maximum at 200 *m*/*z*. The mass range in the full-scan experiments is 50–850 *m*/*z*. The conditions in the positive ionization mode (ESI+) are the following: spray voltage 3500 V; capillary temperature 320 °C; S-lens RF level 50; sheath gas pressure (N_2_ > 95%) 40; auxiliary gas (N_2_ > 95%) 20; and auxiliary gas heater temperature 320 °C. The conditions in the negative ionization mode (ESI−) are the following: spray voltage 2800 V; capillary temperature 320 °C; S-lens RF level 50; sheath gas pressure (N_2_ > 95%) 35; auxiliary gas (N_2_ > 95%) 15; and auxiliary gas heater temperature 320 °C. The parameters for the scan event of PRM are the following: mass resolving power of 17,500 for full width at half maximum (200 *m*/*z*), an AGC target at 2 × 10^5^, a maximum IT at 100 ms, and an isolation window at 2.0 *m*/*z* for accurate mass measurement fragments.

Standards of different mycotoxins (purity > 98%) were purchased from Sigma-Aldrich (St. Louis, MO, USA) and VWR (Radnor, PA, USA). The following mycotoxins were searched (Appendix A): aflatoxin B1, aflatoxin B2, aflatoxin G1, aflatoxin G2, fumonisin B1, fumonisin B2, fumonisin B3, deoxynivalenol, fusaric acid, beauvericin, zearalenone, T-2, HT-2, and OTA. The retention time, electrospray ionization modes, molecular weight, parallel reaction monitoring values in tandem MS/MS, collision energies, and ESI adducts for these mycotoxins are available in Appendix A. In accordance with the guidelines established by the European Union Reference Laboratory and Regulation (EU) 2021/808, calibration curves were prepared in both solvent (acetonitrile/water, 50:50 *v*/*v*) and silage extracts, using five concentration levels ranging from 1 to 100 μg/kg. These were used to assess linearity, expressed as the correlation coefficient (R^2^), and to evaluate the matrix effect (ME). For the latter, the slopes of the calibration curves in the solvent and in the matrix were compared to determine the signal suppression/enhancement factor. The limit of detection (LOD) was calculated based on the standard deviation (σ) of replicate measurements of spiked samples at a low concentration (1 μg/mL), using the slope of the calibration curve generated in silage matrix, according to the following equations: LOD = 3.3 × (σ/slope); LOQ = 10 × (σ/slope). The matrix effect (ME) was assessed by comparing the slopes of the calibration curves prepared in the solvent and in the silage matrix. Recovery was evaluated by spiking silage extracts at three concentration levels of the standard mix: low (1 μg/L), medium (10 μg/L), and high (25 μg/mL). The recovery values ranged from 62.7% to 137%, with associated RSD (%) values between 8% and 19%. Finally, calibration curves in the solvent for FA and BEA were confirmed for linearity by obtaining R^2^ values > 0.99. All relevant validation parameters related to the targeted mycotoxins are summarized in Appendix A.

The untargeted metabolomic profiling was carried out on the same full-scan MS raw data, using the software MS-DIAL (version 4.90) for data elaboration. The mass range of 50–850 *m*/*z* was searched for features with a minimum peak height of 10,000 cps. The MS and MS/MS tolerances for peak centroiding were set to 0.05 and 0.1 Da, respectively. The accurate mass tolerance for identification was 0.05 Da for MS and 0.1 Da for MS/MS. The identification step was based on mass accuracy, the isotopic pattern, and spectral matching. In MS-DIAL, these criteria were used to calculate the total identification score. The total identification score cut-off was >50%, considering the most common ESI+ adducts. Gap filling using the peak finder algorithm was performed to fill in the missing peaks, considering 5 ppm tolerance for *m*/*z* values. The ESI-positive MSMS library of MS-DIAL was coupled with a custom database containing a list of mycotoxins and main metabolites for tentative annotation according to the accurate mass and isotopic profile of each compound.

### 5.4. Statistical Analysis

The multivariate data analysis on the elaborated mass features was conducted using different available software platforms, namely MetaboAnalyst 6.0 and SIMCA 18 (Umetrics, Malmo, Sweden), for both unsupervised and supervised statistical modelling, namely a hierarchical cluster analysis (HCA, Euclidean distance) and an orthogonal projection to latent structure discriminant analysis (OPLS-DA), respectively. In addition,, one-way analysis of variance (ANOVA; *p* < 0.05, Duncan’s post hoc test) was conducted using IBM PASW Statistics 26.0 (SPSS Inc., Chicago, IL, USA) to find significant differences in mycotoxin distribution. Additionally, the “rAMOPLS” package (version 0.2) on R studio (version 4.2.3) was used for ANOVA multi-block orthogonal partial least squares analysis (AMOPLS), to check for the significance of geographical origin, sampling season, and their interaction when considering the untargeted chemical profile of CS and HMC samples. Several statistical parameters, such as goodness of fit (R^2^Y), residual structure ratio (RSR), residual sum of squares (RSS), and their associated *p*-values were evaluated.

## Figures and Tables

**Figure 1 toxins-17-00214-f001:**
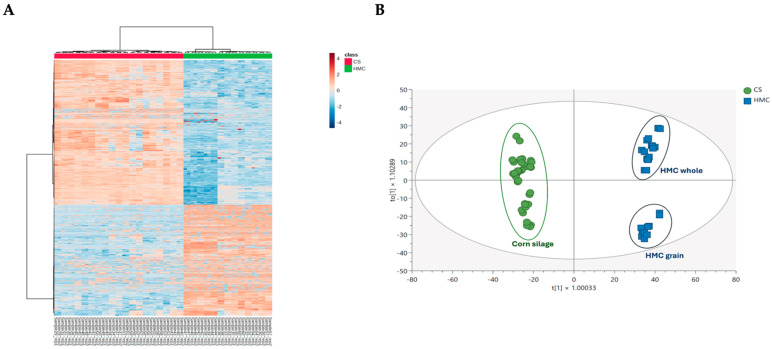
(**A**) Heat map resulting from the unsupervised hierarchical cluster analysis, together with the (**B**) OPLS-DA score plot showing the differences between corn silage (CS) and high-moisture corn (HMC) samples (both as whole and grain formulations). The heat map was built using the normalized dataset (autoscaling each feature), Euclidean as distance measure, and Ward as clustering method. Each colored cell on the map corresponds to a normalized abundance value, with samples in rows and features/compounds in columns.

**Figure 2 toxins-17-00214-f002:**
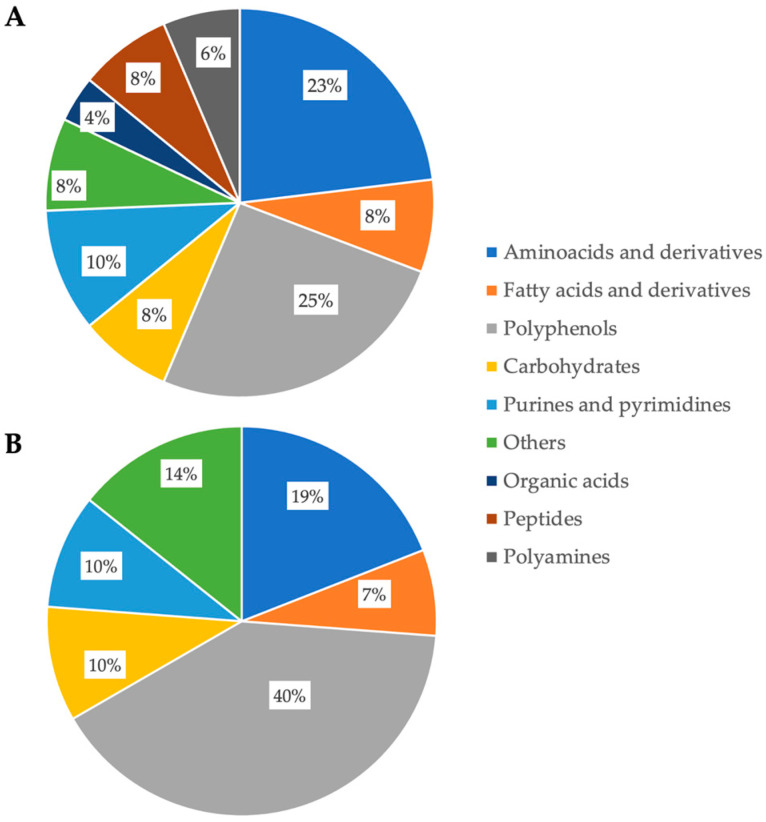
Pie chart showing the chemical diversity of significant metabolites that were related to the discrimination of CS (**A**) vs. HMC (**B**).

**Figure 3 toxins-17-00214-f003:**
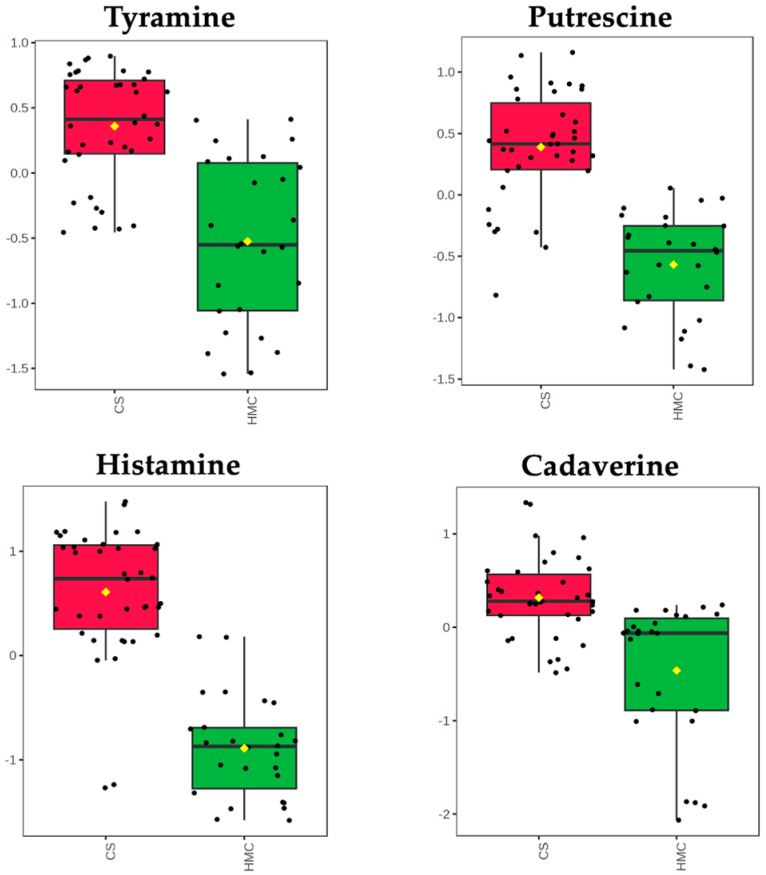
Abundance values (log_10_ transformed, Pareto scaled, and normalized by median, shown on Y axes) outlining significant variations in different polyamines for the pairwise comparison CS vs. HMC and resulting from the volcano plot analysis (MetaboAnalyst 6.0).

**Figure 4 toxins-17-00214-f004:**
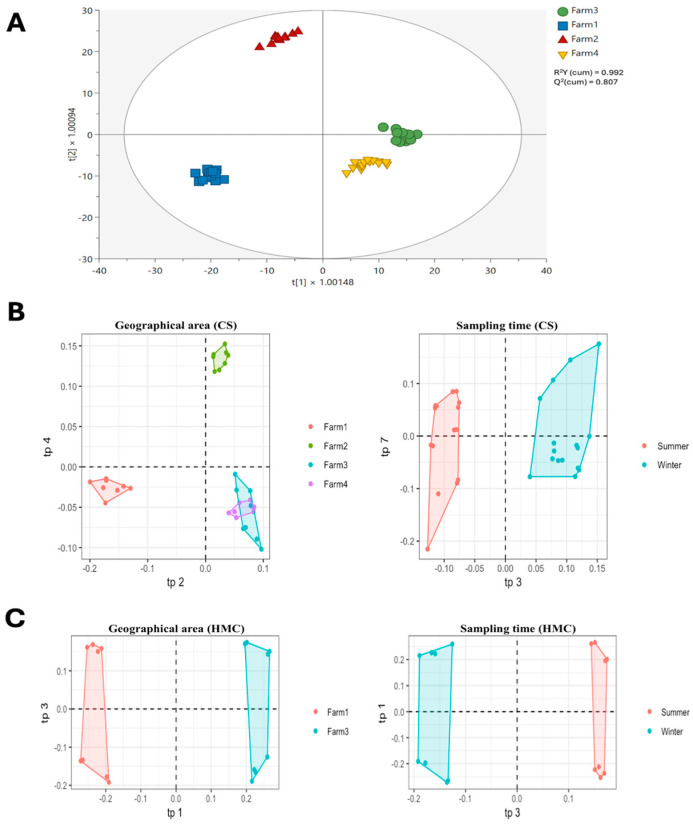
(**A**) OPLS-DA score plot for the discrimination of both CS and HMC samples according to their geographical area; (**B**) score plots on the effect of both geographical area and sampling time for the CS group; (**C**) score plots on the effect of both geographical area and sampling time for the HMC group. For (**A**), t[1] = first latent vector of OPLS-DA score plot; t[2] = second latent vector of OPLS-DA score plot. t[1] and t[2] are extrapolated considering *p* < 0.05 as a significance level, to compute the Hotelling’s T^2^ ellipse and the critical distance to the model. For (**B**,**C**), resulting from AMOPLS analysis, the different axes report contributions to each main predictive component (i.e., tp) able to discriminate the different groups.

**Table 1 toxins-17-00214-t001:** Quantitative values of the most abundant regulated (fumonisin B1, B2, B3, zearalenone, and deoxynivalenol) and emerging (beauvericin and fusaric acid) mycotoxins found in both corn silage (CS) and high-moisture corn (HMC) samples under investigation. Abbreviations: nd = not detected; FB1 = fumonisin B1; FB2+FB3 = cumulative values of fumonisin B2 and B3; ZEN = zearalenone; DON = deoxynivalenol; BEA = beauvericin; FA = fusaric acid. The results are mean values of two (*n* = 2) pooled biological replicates and expressed as μg/kg of dry matter (DM) analyzed.

Group	Sample ID	FB1 (μg/kg)	FB2+FB3 (μg/kg)	ZEN (μg/kg)	DON (μg/kg)	BEA (μg/kg)	FA (μg/kg)
CS (n = 19)	Sample 29	352.9 ± 16.5	156.6 ± 1.9	27.2 ± 1.3	250.2 ± 10.4	17.0 ± 1.4	788.2 ± 5.2
	Sample 31	365.5 ± 13.7	166.4 ± 0.9	19.5 ± 0.5	515.2 ± 15.6	21.1 ± 1.7	906.4 ± 5.7
	Sample 34	371.5 ± 19.0	164.1 ± 2.9	33.2 ± 3.4	684. 5 ± 5.2	4.5 ± 0.6	702.6 ± 6.1
	Sample 36	322.9 ± 1.7	170.0 ± 1.1	19.1 ± 2.6	404.8 ± 15.6	18.5 ± 0.6	817.2 ± 0.7
	Sample 37	315.5 ± 4.3	160.0 ± 0.8	120.0 ± 10.6	1104.0 ± 52.0	10.9 ± 0.3	1040.0 ± 2.8
	Sample 40	722.7 ± 26.8	483.6 ± 4.4	nd	220.8 ± 0.5	14.6 ± 4.3	722.9 ± 6.1
	Sample 41	239.2 ± 4.6	157.5 ± 5.9	10.3 ± 1.3	345.9 ± 10.4	8.2 ± 0.3	205.5 ± 0.6
	Sample 43	410.3 ± 1.5	179.2 ± 1.7	18.7 ± 3.2	132.5 ± 7.3	9.0 ± 0.5	555.9 ± 3.9
	Sample 45	1043.9 ± 19.2	252.2 ± 1.5	30.0 ± 6.4	345.9 ± 12.3	27.9 ± 1.6	1425.4 ± 10.0
	Sample 48	264.0 ± 3.4	162.6 ± 2.6	10.7 ± 0.8	71.4 ± 2.6	8.2 ± 0.6	673.0 ± 6.8
	Sample 49	208.0 ± 9.5	154.7 ± 6.0	3.5 ± 0.3	287.0 ± 20.8	2.3 ± 0.2	504.0 ± 17.4
	Sample 50	304.4 ± 11.6	159.2 ± 4.1	nd	53.7 ± 5.2	4.2 ± 0.2	338.2 ± 7.9
	Sample 51	410.5 ± 28.7	173.0 ± 4.8	5.2 ± 1.5	14.7 ± 4.2	12.8 ± 0.8	513.4 ± 8.9
	Sample 52	904.2 ± 38.1	504.8 ± 8.7	20.4 ± 0.3	956.8 ± 52.0	54.6 ± 1.6	2304.8 ± 0.7
	Sample 53	274.5 ± 0.1	163.5 ± 0.1	11.1 ± 5.5	272.3 ± 10.3	10.6 ± 1.1	646.0 ± 14.0
	Sample 55	1018.8 ± 48.5	569.1 ± 15.3	26.1 ± 5.6	368.0 ± 5.3	47.9 ± 4.7	1565.7 ± 3.5
	Sample 56	368.6 ± 4.2	188.6 ± 4.8	6.4 ± 0.3	44.2 ± 1.1	11.4 ± 0.6	727.5 ± 1.8
	Sample 58	1953.5 ± 89.8	775.2 ± 10.8	15.2 ± 1.3	169.3 ± 9.8	172.7 ± 7.9	3011.3 ± 11.1
	Sample 59	273.9 ± 23.1	171.8 ± 1.9	61.9 ± 2.6	736.0 ± 5.1	11.9 ± 1.1	287.9 ± 0.6
	mean value	533.2	258.5	23.1	367.2	24.7	933.5
HMC (n = 13)	Sample 39	3110.7 ± 58.2	1103.0 ± 2.0	13.3 ± 0.8	95.7 ± 8.2	23.4 ± 1.6	1836.5 ± 0.9
	Sample 42	3063.4 ± 10.8	1191.9 ± 25.8	nd	9.6 ± 1.9	36.8 ± 4.7	4783.5 ± 96.1
	Sample 44	3158.2 ± 34.9	1144.6 ± 23.8	35.4 ± 2.9	23.6 ± 1.0	41.2 ± 1.6	2983.9 ± 24.0
	Sample 46	2383.9 ± 29.5	916.2 ± 7.7	30.6 ± 1.3	36.0 ± 1.1	32.3 ± 1.6	2531.0 ± 12.4
	Sample 47	1858.9 ± 63.2	734.3 ± 17.0	35.6 ± 2.6	28.7 ± 3.1	26.7 ± 3.1	1359.8 ± 8.1
	Sample 30	359.1 ± 16.2	156.4 ± 8.2	412.4 ± 53.0	1658.0 ± 104.1	5.0 ± 0.2	622.1 ± 9.4
	Sample 32	439.2 ± 0.7	174.8 ± 3.3	30.9 ± 0.3	49.3 ± 3.6	14.2 ± 1.4	611.5 ± 3.0
	Sample 33	525.9 ± 17.1	185.4 ± 5.3	27.7 ± 2.6	36.8 ± 2.6	9.6 ± 0.9	387.8 ± 9.8
	Sample 35	434.3 ± 1.6	171.4 ± 2.6	8.8 ± 0.3	31.7 ± 1.1	7.5 ± 0.5	305.8 ± 2.8
	Sample 38	313.8 ± 14.4	166.5 ± 1.8	101.2 ± 5.3	272.3 ± 26.1	2.7 ± 0.2	429.7 ± 13.9
	Sample 54	386.4 ± 8.8	173.5 ± 0.2	238.1 ± 18.5	706.5 ± 10.3	8.8 ± 0.2	391.2 ± 19.0
	Sample 57	431.1 ± 32.0	183.9 ± 6.7	27.9 ± 0.8	55.9 ± 5.2	13.0 ± 2.7	524.6 ± 12.8
	Sample 60	639.3 ± 32.9	198.6 ± 0.7	54.4 ± 2.6	265.0 ± 36.4	16.8 ± 1.7	795.0 ± 2.6
	mean value	1315.7	500.0	78.2	259.2	18.3	1350.9

**Table 2 toxins-17-00214-t002:** Relative variability and block contributions of the AMOPLS analysis of both (CS) and high-moisture corn (HMC) metabolome data acquired considering the three different factors under investigation (i.e., origin, sampling, origin × sampling).

Corn Silage (CS)	RSS (%)	RSR	RSS *p*-Value	RSR *p*-Value	R^2^Y *p*-Value	tp1	tp2	tp3	to1
Origin	20.7%	1.329	0.01	0.01	0.01	83.4%	93.8%	2.8%	20.9%
Sampling	5.4%	1.080	1.00	1.00	0.01	5.4%	2.0%	90.2%	25.7%
Origin × Sampling	10.3%	1.089	1.00	1.00	0.01	5.4%	2.0%	3.4%	25.5%
Residuals	63.7%	1.000	N/A	N/A	N/A	5.9%	2.2%	3.7%	27.8%
**High-Moisture Corn** **(HMC)**	**RSS** **(%)**	**RSR**	**RSS** ***p*-Value**	**RSR** ***p*-Value**	**R^2^Y** ***p*-Value**	**tp1**	**tp2**	**tp3**	**to1**
Origin	29.5%	2.018	0.01	0.01	0.01	98.5%	0.0%	0.5%	15.3%
Sampling	7.8%	1.117	1.00	1.00	0.01	0.5%	0.1%	97.8%	27.7%
Origin × Sampling	9.6%	1.191	1.00	1.00	0.01	0.5%	99.8%	0.8%	26.0%
Residuals	53.2%	1.000	N/A	N/A	N/A	0.05%	0.1%	0.9%	31.0%

Abbreviations: N/A = not available; RSS = relative sum of squares; RSR = residual structure ratio; tp = predictive component; to = orthogonal component.

## Data Availability

The original contributions presented in this study are included in the article; further inquiries can be directed to the corresponding author.

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
