# Peer review of "Evaluation of Untargeted Metabolomic and Mycotoxin Profiles in Corn Silage and High-Moisture Corn"

_toxins, 2025, doi:10.3390/toxins17050214_

Round 1

Reviewer 1 Report

Comments and Suggestions for Authors

  A very interesting study combining the complementary studies of metabolomics and mycotoxin screening to assess feed quality, identify biomarkers to better understand the link between fungal contamination and biochemical composition. 
Additional materials are welcome on metabolomics, microbiological and mycotoxicological examinations, retention times, quantification limits. 
However I consider that the References part should be checked once more and numbered where only authors appear ( line 338, Gauthier et al., 2015). 
I also consider the Discussion chapter a bit long, some paragraphs could maybe go into the introduction and shortened a few.  

Author Response

Reviewer #1

A very interesting study combining the complementary studies of metabolomics and mycotoxin screening to assess feed quality, identify biomarkers to better understand the link between fungal contamination and biochemical composition.

Authors: We would like to thank the reviewer for having appreciated the overall aim of the work.

Additional materials are welcome on metabolomics, microbiological and mycotoxicological examinations, retention times, quantification limits.

Authors: We have added new information about method validations and detection (Table S1, sheet d.) about the mycotoxins tested and quantified. Also, more details about metabolomics profile have been provided by improving Figures captions.

However, I consider that the References part should be checked once more and numbered where only authors appear ( line 338, Gauthier et al., 2015).

Authors: Thank you for checking the Reference style. It has been improved during the revision step.

I also consider the Discussion chapter a bit long, some paragraphs could maybe go into the introduction and shortened a few. 

Authors: The discussion section has been shortened where possible. The length of the discussion section is strictly dependent from the huge amount of significant metabolomics findings to be discussed as related with CS and HMC profiles. Thank you for pointing it out and for the understanding.

Reviewer 2 Report

Comments and Suggestions for Authors

Comments to the Authors  

In the study, the authors deal with the innovative application of untargeted metabolomics combined with mycotoxins to evaluate the chemical profiles and contamination in corn silage and high-moisture corn. Overall, the manuscript is in general well-written in objectives. The materials and method are no major flaw. This document is of informative for analysis, animal welfare and food safety. Suggestions appear below.

Major comments:

  1. The authors did not provide any information about the analytical validation method. To make sure the result is reliable, the method validation should be shown. There are actually some method validation data shown, including LOD and LOQ. Further data is mandatory though, e.g. the recovery of the overall method for each tested mycotoxin, quality assurance and/or etc.

Minor comments:

  1. Introduction: The authors should provide more information on the impact of the metabolic profiles in animals consuming CS and HMC contaminated with mycotoxins.
  2. Figure 1A is not clear for this reviewer. Could you separate the heat map for each sample, as this will make the details more visible? The scale is also unclear. However, if the scale of the figure or graph (Fig 1A and 1B) is not clear, the image quality cannot be improved, please provide the additional explanation below the graph or figure.
  3. Figure 3. Please clearly specify the X and Y axes along with their units. Please make the numbers and labels more clearly.
  4. Table 2. It is recommended to add the explanation for abbreviations below the table.
  5. Figure 4 is unclear. The scale should be made clearer.
  6. L524-542: It should be summarized in the table.
Comments on the Quality of English Language

Good

Author Response

In the study, the authors deal with the innovative application of untargeted metabolomics combined with mycotoxins to evaluate the chemical profiles and contamination in corn silage and high-moisture corn. Overall, the manuscript is in general well-written in objectives. The materials and method are no major flaw. This document is of informative for analysis, animal welfare and food safety. Suggestions appear below.

Authors: We would like to thank the reviewer for having appreciated the overall aim of the work.

Major comments:

1) The authors did not provide any information about the analytical validation method. To make sure the result is reliable, the method validation should be shown. There are actually some method validation data shown, including LOD and LOQ. Further data is mandatory though, e.g. the recovery of the overall method for each tested mycotoxin, quality assurance and/or etc.

Authors: Thank you for your valuable feedback. We acknowledge the importance of method validation to ensure the reliability of analytical results. We would like to clarify that our approach is not a classical targeted analysis but rather a comprehensive screening method that combines untargeted metabolomics with mycotoxin profiling. This methodology provides a broad-spectrum analysis of metabolites, allowing for an in-depth exploration of biochemical alterations, rather than focusing solely on absolute quantification. Untargeted metabolomics offers an unbiased approach to detecting emerging contaminants and metabolic perturbations, making it a valuable tool even in the absence of complete targeted validation. However, we recognize that analytical validation (including recovery studies for each mycotoxin) are essential in these works, therefore we have performed a method validation on the mycotoxins quantified in silage samples, i.e. those represented a safety issue for the animals (as outlined in revised paragraph 5.3 and Table S1 sheet d.). The intention is to broaden the range of targeted mycotoxins, and the method will be evaluated for these additional compounds in a future phase of the screening method development. As for quality assurance, quality control injections were injected to assess retention time shifts and signal stability. Additionally, the reliability of our developed analytical method was verified by comparing the results with those obtained from the same samples analysed by an accredited laboratory, which followed validated procedures and successfully passed proficiency testing. We hope this clarification addresses the concerns raised, and we appreciate your insightful comments.

Minor comments:

1) Introduction: The authors should provide more information on the impact of the metabolic profiles in animals consuming CS and HMC contaminated with mycotoxins.

Authors: Thank you for the information. We have mentioned and cited the review published by Ogunade and co-authors on the role of mycotoxins contaminating silages towards the risk of metabolic impairment for the animal. Also, other references have been added to support the effects, outlining a role in energy metabolism, amino acid metabolism, liver and gut health, and neurotransmitter or hormonal disruptions.

2) Figure 1A is not clear for this reviewer. Could you separate the heat map for each sample, as this will make the details more visible? The scale is also unclear. However, if the scale of the figure or graph (Fig 1A and 1B) is not clear, the image quality cannot be improved, please provide the additional explanation below the graph or figure.

Authors: We understand this comment considering that the heat map is calculated using all the annotated mass features. A heatmap provides intuitive visualization of a data table. Each colored cell on the map corresponds to a concentration value in your data table, with samples in rows and features/compounds in columns. The heat map has been used as unsupervised tool to check the separation between CS and HMC groups, rather than each sample. This was done to provide a first picture of those groups of mass features clearly discriminating corn silage (CS) from high moisture corn (HMC). The same output was confirmed by using a supervised tool, such as OPLS-DA; particularly, it was evident from the OPLS-DA score plot the same separation outlined in the heat map, based on the different metabolomic profile of these feeds under investigation. Therefore, at this level, we are not interested in the effect of geographical origin or sampling season, considering that the significant effect of these factors was then assessed with another statistical approach, namely AMOPLS, thus allowing to evaluate also the interaction between these two factors. As far as the scaling is concerned, the Heat Map was built on the software MetaboAnalyst 6.0, while the OPLS-DA score plot was done on the software SIMCA (Umetrics). For the heat map, the variation of each metabolite is provided referring to the normalization strategy used, namely: Log10 Transformation, Normalization by median, and Pareto Scaling. Regarding OPLS-DA score plot, the X and Y axes refer to the first (t[1]) and second (t0[1]) model components (namely predictive and orthogonal). More information have been provided in the Figure 1 caption.

3) Figure 3. Please clearly specify the X and Y axes along with their units. Please make the numbers and labels more clearly.

Authors: Thank you for the suggestion. In MetaboAnalyst 6.0, the second box plot on the right typically represents the normalized concentration values after applying one of the platform’s normalization methods. Based on common normalization techniques used in metabolomics, the normalization performed could include: Log Transformation (converts data to a log scale to reduce skewness); Normalization by median (adjusts for systematic differences across samples by centering each sample around the median); Pareto Scaling (divides by the square root of the standard deviation instead of the standard deviation). We have added more information in the Figure 3 caption, improving also the resolution. 

4) Table 2. It is recommended to add the explanation for abbreviations below the table.

Authors: Thank you for the suggestion. Abbreviations have been provided below the table.

5) Figure 4 is unclear. The scale should be made clearer.

Authors: Figure 4 has been improved by providing more details about scaling, together with information on x and y axes of each score plot.

6) L524-542: It should be summarized in the table.

Authors: We have added a comprehensive new table (Table S1) summarizing all these details. Thank you for pointing it out.

Reviewer 3 Report

Comments and Suggestions for Authors

Attached on the file "Review Comments_Toxins"

Comments on the Quality of English Language

Written in clear academic English, but still some modifications should be done. 

Author Response

This article “Untargeted Metabolomic and Mycotoxin Profiles in Corn Silage and High-Moisture Corn” explores the chemical complexity and mycotoxin contamination risks associated with corn silage (CS) and high-moisture corn (HMC), two essential components of ruminant diets. Using UHPLC-HRMS-based untargeted metabolomics and mycotoxin profiling, the study characterizes the metabolic and contamination patterns of CS and HMC samples collected from Northern Italy over two years. With a particular focus on feed safety and quality assessment, the article describes how fumonisin B1 (FB1) emerged as the predominant mycotoxin, with significantly higher concentrations in HMC than in CS, though within EU limits. It highlights how untargeted metabolomics effectively distinguishes CS and HMC based on their metabolic signatures, revealing that CS is rich in polyamines, amino acids, peptides, and phenolic acids, while HMC is primarily characterized by flavonoids and mycotoxins. Additionally, the study examines the impact of geographical origin and sampling season on metabolite profiles and mycotoxin distribution, finding that origin plays a significant role, while seasonal variation has a minimal effect. These findings underscore the complementary role of metabolomics and mycotoxin screening in feed quality assessment, biomarker identification, and the evaluation of fungal contamination, offering a robust strategy to enhance feed safety management in livestock production.

Overall, it is well-structured, methodologically sound, and written in clear academic English. But still some modifications should be done. Below, I outline the necessary revisions required for a polished and submission-ready version.

Authors: We would like to thank the reviewer for having appreciated the overall aim of the work. The manuscript has been revised according to each major drawback outlined.

  • Please ensure consistency in terminology. Revise and correct some important words to the field of research (Example: gr- correct to “g”, the unit for mass is grams (g), not grains (gr). The International System of Units (SI) recognizes "g" as the only symbol for gram. Otherwise you are considering “grains”, which corresponds to 0.065 grams).

Authors: We apologize for the misleading term used in paragraph 5.2. The unit for mass has been revised, accordingly.

  • Abbreviations should be defined at first mention and used consistently thereafter. Please verify all (examples: lactic acid bacteria (LAB), corn silage, high-moisture corn …)

Authors: Abbreviations have been checked and revised, accordingly. Thank you for pointing it out.

Here are some areas that could be improved by sections

Remarks

L144: An extra “(“ between “Figure 1” and “A”

Authors: revised, accordingly.

Table 2: there is some information that must be detailed. For example, what does “tp1, tp2, tp3, to1” stand for?

Authors: Table 2 has been improved by adding the "Abbreviations" section.

L338: Include the reference

Authors: revised, accordingly.

L479: Correct the wavelength range

Authors: revised, accordingly.

L492: “acetonitrile”

Authors: revised, accordingly.

Material and methods section:

L526-537: retention time: (RT)(min)

Authors: revised, accordingly. We have introduced a new Table (Table 3), defining all the method-related parameters.

In the results section more data on the method validation can be added.

Authors: We have added more information on method validation as supplementary material (Table S1) and in M&M and Discussion section. Thank you for pointing it out.

Round 2

Reviewer 1 Report

Comments and Suggestions for Authors

I have no other comments, the paper could be published. 

Reviewer 2 Report

Comments and Suggestions for Authors

The revised manuscript has been improved for publication in its current form.